# Systematic Investigation of the Effect of *Lactobacillus acidophilus* TW01 on Potential Prevention of Particulate Matter (PM)2.5-Induced Damage Using a Novel *In Vitro* Platform

**DOI:** 10.3390/foods12173278

**Published:** 2023-09-01

**Authors:** Sioumin Luo, Mingju Chen

**Affiliations:** Department of Animal Science and Technology, National Taiwan University, Taipei 10617, Taiwan, China; siouminluo@ntu.edu.tw

**Keywords:** *Lactobacillus acidophilus*, antioxidants, intestinal protection, immune regulation, anti-CSE injury

## Abstract

Exposure to ambient particulate matter (PM) and cigarette smoking (CS) is a risk factor for respiratory/lung infections and metabolic disorders. Lung–gut axis disruption involving the upregulation of oxidative stress, systemic inflammation, and gut barrier dysfunction by PM is one of the potential mechanisms. Thus, we designed a novel *in vitro* platform for pre-selecting probiotics with potentially protective effects against PM-induced lung damage through the lung–gut axis to reduce animal usage. The results showed that a high dose of *Lactobacillus acidophilus* TW01 (1 × 10^8^ CFU/mL) inhibited reactive oxygen species (ROS) production. This strain could also reduce respiratory epithelial cell death induced by cigarette smoke extraction (CSE), as well as promoting Caco-2 cell migration in 1 × 10^6^ CFU/mL. Although further animal experiments are needed to validate the *in vitro* findings, *L. acidophilus* TW01 is a promising probiotic strain for the potential prevention of PM2.5-induced damage.

## 1. Introduction

Recently, the increasing health risks from air pollution, such as particulate matter (PM), have received increasing attention [1,2]. Fine particulate matter, known as PM2.5 (PM diameter ≤ 2.5 µm), is an air pollutant that is a severe threat to human health worldwide [3]. PM exposure induces several respiratory diseases by prompting pulmonary inflammation and inducing oxidative stress [4,5]. These particles contain lots of dangerous chemical compounds, such as polycyclic aromatic hydrocarbons (PAHs), aromatic ketones, and ethylene glycol, which are easily breathed into the lungs, contributing to many hazardous effects on the airway system [6]. Short-term exposure of PM2.5 significantly increases inflammatory factors in the lung [7]. The World Health Organization warns that the urban population has been exposed to different degrees of PM2.5. PM2.5 is also a major component of cigarette smoke (CS), demonstrating a strong correlation with chronic lung diseases and cancer [3,8,9].

A study in children with cystic fibrosis observed that some bacteria were found in the intestinal tract before being identified in the respiratory tract, suggesting that microaspiration might involve intestinal microbes in the development of respiratory tract microbiota [10,11]. Accumulating evidence also demonstrated that the microbial composition in the intestinal and respiratory tracts was closely related, and an alteration in the microbiota in the intestinal or respiratory tracts could influence the other [11]. Additionally, many pulmonary diseases have been reported as being related to a dysbiosis in the airway and intestinal microbiota, indicating that the two components of the“lung–gut axis” influence each other [12]. PM-mediated microbiota dysbiosis and metabolic disorder may be due to lung–gut axis disruption [13], as PM upregulates lung oxidative stress, systemic inflammation, gut barrier dysfunction, and microbiota dysbiosis [14]. Inflammatory lung diseases (such as asthma, pulmonary emphysema, and even lung cancer) [5,15] are associated with microbiota dysbiosis mediated by PM [16], resulting in impaired gut barrier function and reduced serum lipopolysaccharide (LPS) levels [17].

Probiotics possess important health-promoting characteristics, including maintaining intestinal homeostasis and reducing inflammation; thus, they may provide a possible strategy for preventing PM damage [18,19]. Several probiotic supplements of *Lactobacillus* strains reduce respiratory hyperresponsiveness caused by chronic PM2.5 inhalation [13]. *Lactobacillus acidophilus* has been reported to stimulate and modulate the respiratory immune system in mice [20]. *Lactobacillus paracasei* suppressed PM2.5-induced inflammation in a mouse allergic airway model [19]. *Lactobacillus plantarum* reduced inflammation, blood pressure, and lipid oxidation in smokers [21,22]. The beneficial probiotic protective effects on PM-mediated lung inflammatory diseases [1] might be due to preventing gut leakage, reducing oxidative stress, promoting T-regulatory (Treg) cell progress, and Th1/Th2 balance [9].

Animal welfare is a key issue for conducting animal studies, and the principles of replacement, reduction, and refinement (3Rs) must be adhered to [23]. Thus, in the present study, we designed a novel *in vitro* platform for pre-selecting probiotics with a potential protective effect on PM-induced lung damage through the gut–lung axis. The platform determined the antioxidant ability, immune regulatory effect, preventive effect of cigarette-smoke-induced pulmonary injury, and enhancement of gut barrier.

## 2. Materials and Methods

### 2.1. Preparation of Bacteria

Potential probiotic microorganisms *L. acidophilus* TW01 and *L. paracasei* APL082 were isolated from coffee fermentation and unpasteurized milk, respectively. *L. plantarum* MFM 30-3 and *L. paracasei* MFM 18 were isolated from Mongolian fermented milk (MFM), previously, in Animal Product Lab, National Taiwan University [24]. All strains were inoculated into Lactobacilli MRS broth (Difco Laboratories, Detroit, MI, USA) for 24 h under an anaerobic environment with different temperatures. *L. acidophilus* TW01 and *L. paracasei* APL082 were incubated at 37 °C, whereas *L. plantarum* MFM 30-3 and *L. paracasei* MFM 18 were cultured at 30 °C. The lactic acid bacteria (LAB) were washed with 0.85% sodium chloride (NaCl, Sigma-Aldrich Chemical Co., St. Louis, MO, USA) solution or Dulbecco’s phosphate-buffered saline (DPBS, Gibco, Grand Island, NY, USA) buffer twice before use and suspended in 0.85% NaCl or DPBS buffer at a concentration of 1 × 10^4–8^ CFU/mL for use in different assays. For heat-killed groups, the bacteria (1 × 10^5–6^ CFU/mL density) were bathed at 80 °C for 30 min.

### 2.2. Antioxidant Screening Assays

Radical scavenging activity: The radical scavenging activities were assessed according to the method of Garcia et al. [25]. Each strain was suspended at a density of 10^6–8^ CFU/mL in 0.8 mL DPBS and reacted with 1 mM 1-diphenyl-2-picrylhydrazyl (DPPH, Sigma-Aldrich Chemical Co.) solution (dissolved in ethanol) for 30 min in the dark at room temperature. DPPH solution with 0.8 mL of deionized water was used as the control. All samples were centrifuged at 3000 rpm for 10 min. The absorbance (OD) was measured at 517 nm using a microplate reader (BioTek Epoch, Santa Clara, CA, USA), and the butylated hydroxytoluene (BHT) was dissolved in ethanol under several concentrations (20, 40, 80, 120, 160, and 200 μg/mL) as positive control (Appendix A). The ability of scavenged DPPH was defined as follows:Scavenging activity (%) = (1 − OD_Sample_/OD_blank_) × 100

Ferrozine ion chelating activity: The assay was performed according to the method described by Yusof et al. [26]. Briefly, 10^6–8^ CFU/mL LAB were suspended in 75 μL of 0.85% NaCl solution and mixed with 25 μL of 2 mM ferrous chloride (FeCl_2_, Sigma-Aldrich Chemical Co.) and 50 μL methanol for 30 s. The mixture was reacted with 50 μL of 5 mM ferrozine (Sigma-Aldrich Chemical Co.) solution at room temperature for 10 min. The absorbance was then measured at 562 nm. Ethylenediaminetetraacetic acid (EDTA), which was dissolved in pure water in 3.91, 7.81, 15.63, 31.25, 62.50, and 125 μg/mL, was used as positive control in ferrozine ion chelating activity test (Appendix A). The percentage chelating activity was calculated as follows:chelating activity (%) = (1 − OD_Sample_/OD_blank_) × 100

Thiobarbituric acid reacting substances (TBARS) assay: The catalysis of oxidation used an Fe/H_2_O_2_ system, while linoleic acid served as the source of unsaturated fatty acid. The inhibition of linoleic acid peroxidation was detected using the TBARS method described by Aguilar Diaz De Leon et al. [27]. In brief, LAB strains were suspended at a density of 10^6–8^ CFU/mL in 0.4 mL of 0.85% NaCl solution and mixed with a solution containing 0.2 mL of 0.02 M PBS (Bioman, NTPC, Taiwan), 1 mL linoleic acid emulsification (Sigma-Aldrich Chemical Co.), 0.2 mL 0.01% ferrous sulfate (FeSO_4_, Sigma-Aldrich Chemical Co.), and 0.2 mL 0.02% hydrogen peroxide (H_2_O_2_, Sigma-Aldrich Chemical Co.) in the dark at 37 °C for 12 h. Then, 0.2 mL of 4% thiobarbituric acid (TCA, J.T.Baker, Phillipsberg, NJ, USA), 2 mL of 0.8% thiobarbituric acid (TBA, Sigma-Aldrich Chemical Co.), and 0.2 mL of 0.4% butylated hydroxytoluene (BHT, Sigma-Aldrich Chemical Co.) were added, and the samples were boiled at 100 °C for 30 min. Once the samples had cooled, the absorbance was measured at 532–535 nm. The positive control was BHT (200 μg/mL). ROS inhibition rate was calculated as:ROS inhibition (%) = (1 − OD_Sample_/OD_blank_) × 100

### 2.3. Cell Culture

Caco-2 and Raw264.7 cell lines, purchased from Bioresource Collection and the Research Center (BCRB; Hsinchu, Taiwan), were cultured in Dulbecco’s Modified Eagle Medium (DMEM, Corning, NY, USA) containing 10% inactive Fetal Bovine Serum (FBS, Corning) and 1 mM pyruvate (Corning). Caco-2 cells were supplemented with 0.01 mg/mL human transferrin (Sigma-Aldrich Chemical Co.). The human bronchial epithelial (HBEpiC, Cat:3210) cell line was purchased from ScienCell (Carlsbad, CA, USA) and maintained in Bronchial Epithelial Cell Medium (BEpiCM, Cat:3211, ScienCell). All cell lines were cultured in a 10 cm culture dish (Corning) in a humidified incubator at 37 °C with 5% CO_2_. The cells were harvested using 0.25% Trypsin/EDTA (Corning) when they reached 85−90% confluence.

### 2.4. Co-Culture Model

The co-culture *in vitro* lung model comprised bronchial epithelial HBEpiC and intestinal epithelial Caco-2 cells. Caco-2 cells were seeded onto 3 μm transwell inserts, and fresh medium was added every 2 days for 28 days, in both apical and basolateral sites. The transepithelial electrical resistance (TEER) was measured using an Epithelial Voltoh meter EVOM2 with an STX2 probe according to the manufacturer’s instructions (World Precision Instruments, Sarasota, FL, USA) until the TEER was greater than 350 Ω/cm^2^. The HBEpiC cells were seeded (1.5 × 10^5^ cells/well) into basolateral wells and incubated overnight at 37 °C/5% CO_2_ and then treated with different CSE concentrations. *L. acidophilus* TW01 was added to the apical site for 24 h at the same time.

### 2.5. Wound Healing Migration Assay

This method was adapted from Luo et al. [28]. The 2-well migration inserts (ibidi, DE-BY, Gräfelfing, Germany) were seeded 1 × 10^5^ cells/well Caco-2 cells overnight in 12-well plates. When cell confluence reached 90%, the inserts were removed to make a wound area. The cells were then treated with 1 × 10^4–6^ CFU/mL of *L. acidophilus* TW01 for 16 and 24 h. After treatment, the supernatant was removed, and non-adherent cells were washed off with DPBS buffer. Photographs were taken at 0, 16, and 24 h to calculate the migration rates according to the change in wound area using ImageJ software 1.53k (National Institutes of Health, Bethesda, MD, USA).

### 2.6. Cigarette Smoke Extract (CSE) Preparation

The preparation method of CSE was according to Cheng et al. [29]. Briefly, one cigarette (containing 0.9 mg nicotine and 10 mg tar) was lit. The cigarette smoke was filtered into a 20 mL of the serum-free BEpiCM using a pump until the smoke was completely dissolved in a medium for extraction. The CSE solution was designated as 100% concentration and further diluted with BEpiCM for the experiments. This CSE solution was used no more than 30 min after being filtered through a 0.22 µM filter for each experiment.

### 2.7. MTT Assay

According to the method mentioned in Kumar et al. [30], Raw264.7 cells were seeded into transparent 12-well plates at a density of 10^5^ cells/well and incubated in a humidified 37 °C incubator with 5% CO_2_. The cells were stimulated with different concentrations of *L. acidophilus* with/without 50 ng/mL LPS for 24 h. 3-(4,5-Dimethyl-2-thiazolyl)-2,5-diphenyl-2H-tetrazoliumbromids (MTT, Sigma-Aldrich Chemical Co.) was dissolved in PBS buffer and filtered through a 0.22 µm filter. The conditioned medium was collected to detect cytokines. The MTT reagent was diluted in DMEM and added to each well at final concentration of 0.5 mg/mL. After 4 h incubation at 37 °C kept from light, formazan was dissolved with dimethyl sulfoxide (DMSO, Sigma-Aldrich Chemical Co.), and the absorbance was measured at 560 nm.

### 2.8. Enzyme-Linked Immunosorbent Assay (ELISA)

Raw264.7 cells were seeded at a density of 1 × 10^5^ cells/well and treated with different dosages of bacteria with/without 50 ng/mL LPS for 24 h in 12 well plates. The conditioned medium was collected and centrifuged for 10 min at 4 °C, 12,000 rpm before storage at −80 °C. The cytokines interleukin (IL)-1β, IL-6, IL-10, and tumor necrosis factor (TNF)-α were quantified using ELISA (R&D Systems, Minneapolis, MN, USA) according to the manufacturer’s instructions. The OD was measured on an ELISA reader (BioTek Epoch, Santa Clara, CA, USA) at a wavelength of 450 nm in triplicate.

### 2.9. Cell Cycle Analysis

The method of cell cycle was according to Luo et al. [31]. HBEpiC cells were fixed with 70% cold ethanol and kept at least overnight at −20 °C. The fixed cells were then washed twice with cold DPBS containing 1% FBS. Subsequently, the cells were stained with propidium iodide (PI; Sigma-Aldrich Chemical Co.) solution (50 mg/mL PI in PBS, 1% Tween 20, and 10 mg/mL RNase A) for 45 min at 37 °C in the dark. The DNA content was measured via flow cytometry (FC500, Beckman Coulter Inc., Brea, CA, USA).

### 2.10. Cytometric Bead Array (CBA)

Secreted cytokines in the medium of the CSE-treated HBEpiC were determined using the human Th1/Th2/Th17 CBA assay kit (BD, Franklin Lakes, NJ, USA) via flow cytometry according to the manufacturer’s instructions.

### 2.11. Statistical Analysis

The data are presented as the mean ± standard deviation/structural equation modeling of triplicates, were displayed as percentages with control values, and were compared using one-way analysis of variance (ANOVA) and Tukey’s test. The significant difference was regarded as * *p* < 0.05, ** *p* < 0.01, and *** *p* < 0.001 with negative control, and ^#^
*p* < 0.05, ^##^
*p* < 0.01, and ^###^
*p* < 0.001 with positive control in each experiment.

## 3. Results

### 3.1. L. acidophilus TW01 Presented Better Antioxidant Ability

First, antioxidant ability was used to select the probiotics with potential protection of PM2.5-induced lung damage. For free radical reducing ability (Figure 1A), the DPPH scavenging rate at 10^6^ CFU/mL of *L. acidophilus* TW01, L. plantarum MFM 30-3, *L. paracasei* MFM 18, and *L. paracasei* APL082 was 28.80% ± 8.66%, 46.88% ± 5.62%, 36.72% ± 10.23%, and 12.04% ± 12.04%, respectively. For 10^6^ CFU/mL, the DPPH scavenging rate of *L. acidophilus* TW01, L. plantarum MFM 30-3, *L. paracasei* MFM 18, and *L. paracasei* APL082 was 38.31% ± 13.29%, 24.73% ± 2.49%, 12.22% ± 12.22%, and 23.91% ± 13.08%, respectively. *L. acidophilus* TW01, L. plantarum MFM 30-3, and *L. paracasei* MFM 18 showed a higher DPPH scavenging rate at 10^6–7^ CFU/mL than *L. paracasei*. For the TBARS assay, *L. acidophilus* TW01 at 10^8^ CFU/mL demonstrated a better lipid peroxidation reduction (36.03% ± 18.02%) than *L. plantarum* MFM 30-3 (10.64% ± 6.21%), *L. paracasei* MFM 18 (2.72% ± 2.72%), and *L. paracasei* APL082 (13.81% ± 9.77%) (Figure 1B). However, four tested probiotic strains did not possess ion chelating ability, as measured through Fe^2+^ chelating assay (Figure 1C). According to the results of the TBARS assay and DPPH scavenging rate, *L. acidophilus* TW01 was selected for the following test.

### 3.2. The Immune Regulation of L. acidophilus TW01 Balanced the Th1 and Th2 Immune Response

The MTT assay revealed that *L. acidophilus* TW01 did not damage RAW264.7 cells (Figure 2A). The expression of pro-inflammatory TNF-α and T-reg IL-10 cytokines was significantly upregulated after live *L. acidophilus* TW01 treatment w/wo 50 ng/mL LPS in Raw264.7 cells (Figure 2B,C), whereas significant expression of TNF-α and IL-10 was only observed in heat-killed *L. acidophilus* TW01-treated Raw264.7. For inflammatory cytokines IL-6 and IL-1β, no stimulating effect was found after treatment of live and heat-killed *L. acidophilus* TW01 of Raw264.7 cells. However, both high-dose (10^6^ CFU/mL) living and heat-killed *L. acidophilus* TW01 with 50 ng/mL LPS suppressed the expression of IL-6 and IL-1β in Raw264.7 cells (Figure 2D,E).

### 3.3. L. acidophilus TW01 Protected against CSE-Induced Lung Injury

A novel Caco-2 cell transwell system with HBEpiC cells was developed to investigate the protective effects of this strain on CSE-induced lung damage. The morphology of *L. acidophilus* TW01 on CSE-treated HBEpiC cells (Figure 3A) demonstrated that 20% CSE induced morphology changes in HBEpiC cells. High-dose (10^6^ CFU/mL) *L. acidophilus* TW01 significantly increased cell viability in 20% CSE-treated HBEpiC cells (Figure 3B). We further analyzed the effect of *L. acidophilus* TW01 on the CSE-treated HBEpiC cell cycle (Figure 3C,D), showing significant upregulation of the sub-G1 phase (* *p* = 0.0135), which was significantly suppressed by *L. acidophilus* TW01 (^##^
*p* = 0.0026). TNF-α and IL-6 (Figure 4A,B) were both suppressed by CSE in a dose-dependent manner. High-dose *L. acidophilus* TW01 restored the suppression of both cytokines in CSE-treated HBEpiC cells.

### 3.4. L. acidophilus TW01 Has a Protective Effect on Intestinal Cells

We then determined the intestinal protective effect of *L. acidophilus* TW01, showing that *L. acidophilus* TW01 significantly suppressed Caco-2 cell viability in 10^6–8^ CFU/mL (* *p* = 0.0231, *** *p* < 0.0001, and *** *p* < 0.0001, respectively) (Figure 5) due to acid production. Therefore, low doses of *L. acidophilus* TW01 (10^4–6^ CFU/mL) were selected for the wound recovery assessments to prevent cell cytotoxicity from affecting the outcomes. The cell migration results indicated that *L. acidophilus* TW01 showed a trend of enhancing wound recovery by inducing Caco-2 cell proliferation and migration in 10^6^ CFU/mL after co-culturing for 16 and 24 h (Figure 6A–D).

## 4. Discussion

In most PM2.5 *in vitro* studies, only one or two approaches were included, and the tested cells, such as lung cells, were directly co-cultured with PM2.5 and treated samples, such as drugs [17,32,33]. For studies involving probiotics, the mouse model was usually conducted without an *in vitro* pre-screening test [34]. Investigating the protective effect of probiotics on PM2.5-induced lung damage without a systematic *in vitro* pre-screening test could increase the amount of animal usage during an *in vivo* study. In addition, the lung cells directly co-cultured with PM2.5 and probiotics could not reflect the real situation among probiotics, intestinal cells, and lung cells, illustrating the importance of developing a new approach. Thus, in the present study, we designed an *in vitro* platform for pre-selecting probiotics with a potential protective effect on PM-induced lung damage through determining the antioxidant ability, immune regulatory effect, preventive effect of cigarette-smoke-induced pulmonary injury, and enhancement of the gut barrier, which includes most of the damage factors induced by PM2.5. A Caco-2 cell transwell system with HBEpiC cells was also developed to mimic the probiotic introducing pathway.

After preliminary antioxidative screening, *L. acidophilus* TW01 was selected due to possessing a free radical reducing ability in the DPPH scavenging rate and lipid peroxidation reduction. Exposure to particles in moderate concentrations has been reported to induce oxidative stress [26]. The initial phase of the pulmonary response to PM 2.5 exposure was related to metal ions (Al, As, Cr, Cu, and Zn), which is described as intensely associated with the production of oxidative stress [35] by producing reactive oxygen species (ROS) and assisting superoxide anions (O^2−^) and hydrogen peroxide (H_2_O_2_) conversion to hydroxyl ions (OH^−^) [35,36]. In addition to metal ions, certain organic components, such as polycyclic aromatic hydrocarbons, in soluble fractions and a change in mitochondrial function/NADPH-oxidase can also generate ROS and reactive nitrogen species [37]. The resultant pulmonary effects would include surfactant dysfunction [38], epithelial damage, increased vascular permeability, and inflammatory response, followed by impaired pulmonary function [39,40]. Many studies showed that lactic acid bacteria could produce antioxidative enzymes to defend ROS, including *L. acidophilus* [35,41,42]. *L. acidophilus* has been reported to inhibit the cytotoxicity of 4-nitroquinoline-1-oxide (4NQO) and other oxidants to intestine cells [43]. In the current study, *L. acidophilus* TW01 also demonstrated the ability of reduced free radicals and super-oxidants.

Free radicals and oxidative stress are widely involved in the inflammatory response associated with exposure to PM 2.5 [35,36]. A previous study indicated that, after being treated with PM2.5, the phagocytosis of Raw 264.7 cells was observed, followed by inflammation triggered by the the release of monocyte chemoattractant protein-1(MCP-1), tumor necrosis factor-α (TNF-α), and interleukin-6 (IL-6) and upregulated mRNA expression of inducible nitric oxide synthase (iNOS) [37]. Thus, evaluating the immune regulatory property is also essential for selecting probiotics with attenuation of PM2.5-induced lung damage. Probiotics have been reported to mitigate allergies and respiratory diseases induced by PM2.5 by inducing various cytokines, immune cells, and immunoglobulins in animal models [34]. Our result is similar to prior studies. *L. acidophilus* TW01 demonstrated an anti-inflammatory effect by suppressing inflammatory-related cytokines IL-1β, IL-6, and TNF-α *in vitro*, suggesting the potential ability for the attenuation of PM2.5-induced lung damage.

Cigarette smoke extract (CSE), an airborne PM, has been known to increase cell transformation on epigenetic and genetic factors through different mechanisms, altering the cell death process and favoring inflammation and malignancy, leading to lung injury and tumorigenesis [38]. Thus, CSE was selected as a PM2.5 to evaluate the protective effect of *L. acidophilus* TW01 against PM2.5-induced lung injury. In the present study, we demonstrated that a high dose (10^6^ CFU/mL) of *L. acidophilus* TW01 could significantly increase the cell viability induced by CSE in a Caco-2 cell transwell system with HBEpiC cells, suggesting a protective effect of *L. acidophilus* TW01 on CSE-induced cell death. Apoptotic cells are recognized either on the basis of their reduced DNA-associated fluorescence as cells with diminished DNA content (sub-G1) or morphologic changes [39]. CSE induced mitochondria and nucleus DNA damage in human endothelial cells [40] through inducing p21 cip, leading to cell cycle arrest in the sub-G1 phase [41], followed by initiation of apoptosis. *L. acidophilus* TW01 significantly reduced the cell cycle in the sub-G1 phase, indicating its potential protective effect on CSE-induced cell death. However, chemical components in PM2.5, originating from various sources with diverse composition [44], are the key factor affecting adverse health effects [45]. The source of PM2.5 used in the study could significantly affect the results.

Additionally, ingested PM could increase ROS production and damage the gut epithelial cells through generating toxic metabolites by resident gut microbes, leading to inducing gastrointestinal inflammation and enhancing gut leakage [42]. *L. acidophilus* TW01 showed a trend to recover the injured intestinal cell, suggesting a gut-barrier-protecting effect. This finding is in line with other studies. Various *L. acidophilus* strains were reported to exert a beneficial effect on the intestinal epithelial monolayer [46,47]. Caco-2 cells, a well-published cell line, are mainly used as an intestinal epithelial barrier model [43]. However, we noticed that a high dose of *L. acidophilus* TW01 with high acid production could damage the single layer of attached caco-2 cells. In the intestinal barrier, tight-junction proteins provide an important physical barrier [48]. The low cell viability might be due to the cytotoxicity of lactic acid bacteria [49], resulting in a difficulty for Caco-2 cells in resisting the organic acids produced by *L. acidophilus* TW01.

## 5. Conclusions

In the present study, we systematically screened probiotics on the potential prevention of CSE-induced damage by a novel *in vitro* platform. One strain, *L. acidophilus* TW01, possessed a potential protective effect on CSE-induced cell damage and death *in vitro*. A reduction in cell death by *L. acidophilus* TW01 might be involved in preventing cell damage due to antioxidative and immune regulatory effects, leading to reduced sub-G1 phase. *L. acidophilus* TW01 also showed a trend to recover the injured intestinal cell, suggesting a gut-barrier-protecting effect. This study not only provides a novel screening platform for pre-screening potential probiotics in the prevention of PM2.5-induced damage, as it also illustrates the feasibility of this novel platform by successfully selecting a probiotic strain, *L. acidophilus* TW01, with respiratory protective potential. Further *in vivo* studies are necessary to verify the *in vitro* finding.

## Figures and Tables

**Figure 1 foods-12-03278-f001:**
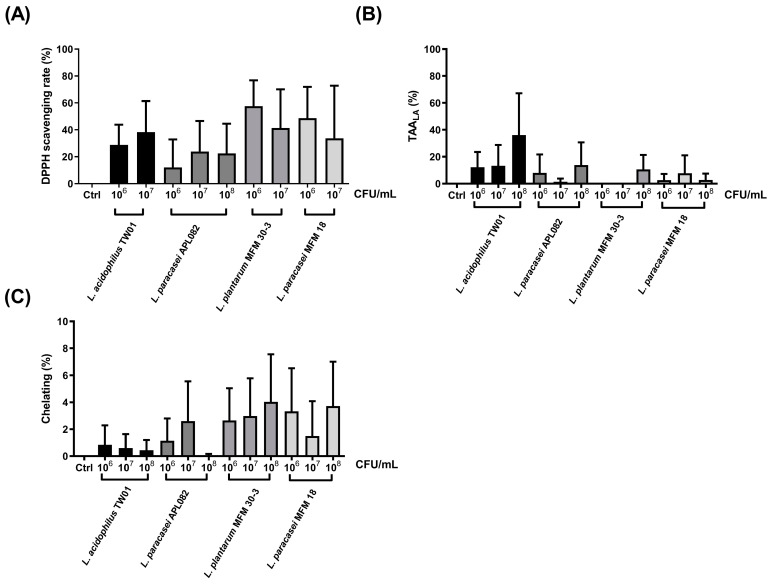
The antioxidant abilities of four LAB: (**A**) the DPPH assay, (**B**) the TBARS assay, and (**C**) the ferrozine assay. The data are presented as mean ± SEM (n = 3). Columns marked with the same colors represented the same strain with different dosages.

**Figure 2 foods-12-03278-f002:**
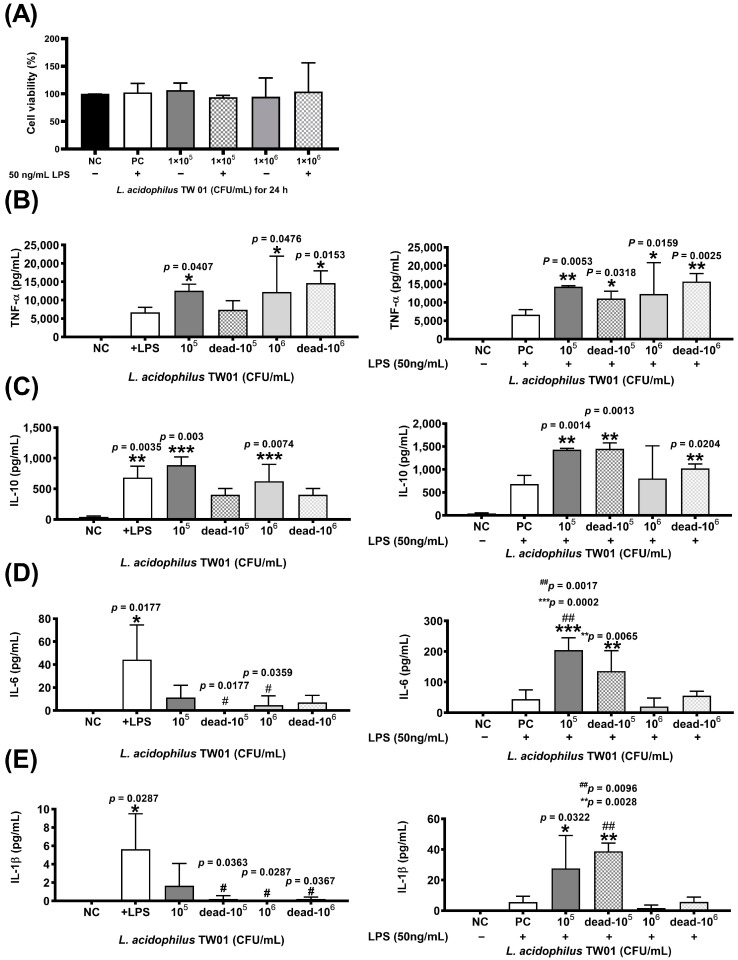
The effect of *L. acidophilus* TW01 on immune regulation. (**A**) Cell viability with or without LPS of *L. acidophilus* TW01-treated Raw264.7 cells at 24 h. Cytokine production of stimulated RAW264.7 cells in response to live or heal-killed *L. acidophilus* TW01 with or without 50 ng/mL LPS. (**B**–**E**) The concentration of TNF-α, IL-10, IL-6, and IL-1β in the conditioned cell media. The data are presented as mean ±SEM (n = 3). The symbols indicate a significant difference compared to the negative (* *p* < 0.05, ** *p* < 0.01, *** *p* < 0.001) and +LPS/positive (^#^
*p* < 0.05, ^##^
*p* < 0.01) controls.

**Figure 3 foods-12-03278-f003:**
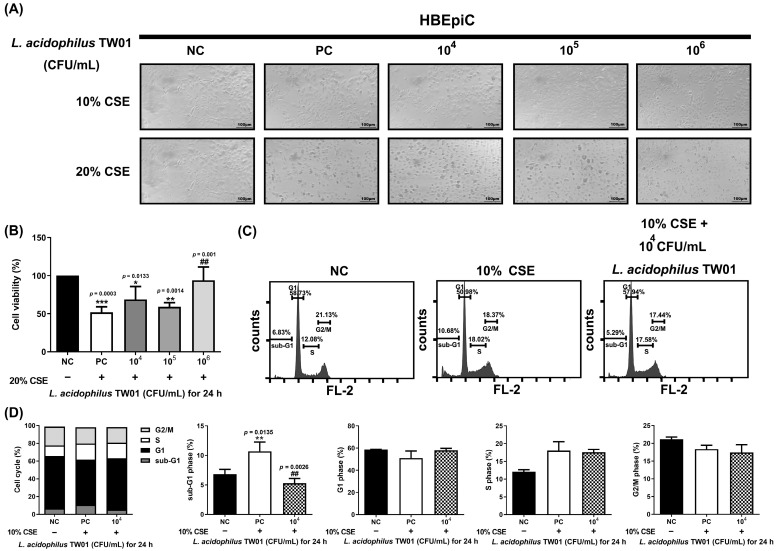
The effect of *L. acidophilus* TW01 on CSE-treated HBEpiC cells. (**A**) 100× morphology of CSE-treated HBEpiC cells at 24 h. (**B**) Cell survival of 20% CSE-treated HBEpiC cells. (**C**) Effect of *L. acidophilus* TW01 on cell cycle regulation in CSE-treated HBEpiC cells. (**D**) Quantitative analysis of the cell cycle data. The data are presented as mean ± SD (n = 3). The symbols indicate a significant difference compared to the negative (* *p* < 0.05, ** *p* < 0.01, and *** *p* < 0.001) and positive (^##^
*p* < 0.01) controls.

**Figure 4 foods-12-03278-f004:**
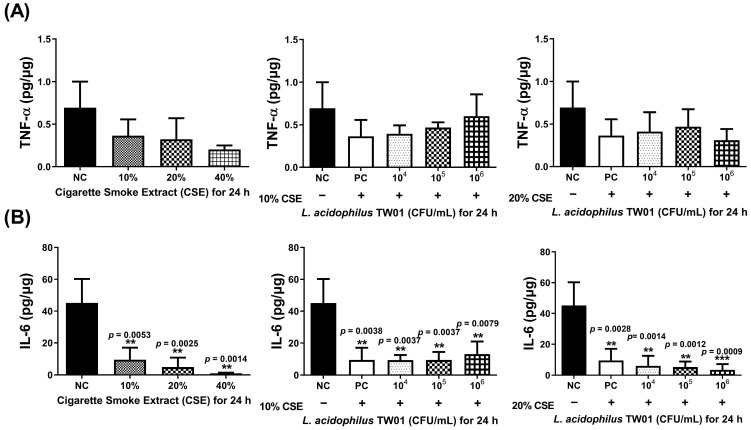
The effect of *L. acidophilus* TW01 on cytokine production of CSE-treated HBEpiC cells for 24 h. (**A**) TNF-α production of CSE-treated HBEpiC cells with and without *L. acidophilus* TW01 for 24 h. (**B**) IL-6 concentration in CSE-treated HBEpiC cells with and without *L. acidophilus* TW01 for 24 h. The data are presented as mean ± SD (n = 3). The symbols indicate a significant difference compared to the negative control (** *p* < 0.01, and *** *p* < 0.001).

**Figure 5 foods-12-03278-f005:**
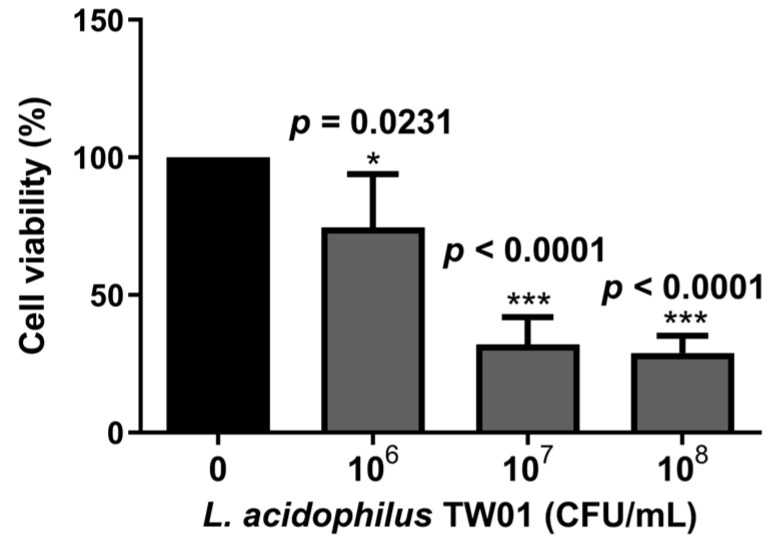
The survival effect of *L. acidophilus* TW01 on Caco-2 cells for 24 h. Data are presented as mean ± SD (n = 3). The symbols indicate a significant difference compared to the negative control (* *p* < 0.05, and *** *p* < 0.001).

**Figure 6 foods-12-03278-f006:**
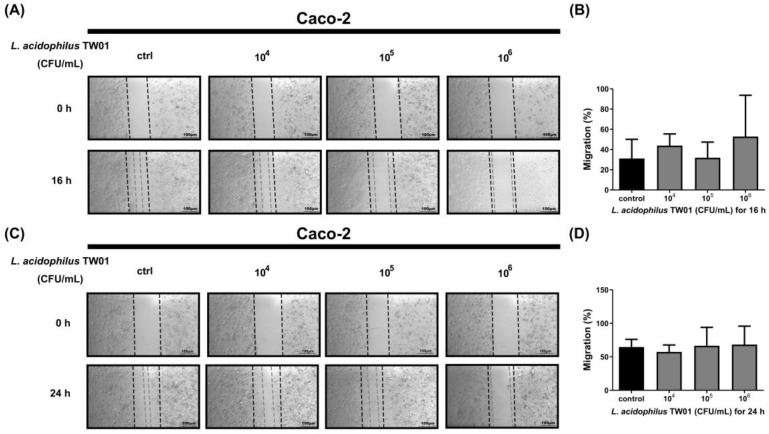
The migration effect of *L. acidophilus* TW01 on Caco-2 cells for 16 h (**A**)and 24 h (**C**). The wound area was photographed under a microscope at 100×. The empty area was quantified with Image J 1.53k (**B**,**D**).

## Data Availability

Data are contained within the article.

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
