# Peer review of "Systematic Investigation of the Effect of Lactobacillus acidophilus TW01 on Potential Prevention of Particulate Matter (PM)2.5-Induced Damage Using a Novel In Vitro Platform"

_foods, 2023, doi:10.3390/foods12173278_

Round 1

Reviewer 1 Report

The manuscript describes the in vitro protective effect of L. acidophilus on particulate matter induced damage. The manuscript has several interesting features, and it should maker minor revision.

Introduction section: some related recent published articles must be added into the revised manuscript.

Discussion section: It is recommended to merge Results and Discussion sections to better illustrate the mechanisms of action involved in this protective effect. Some subtitles of the results have no corresponding discussion in this manuscript. Furthermore, the citation of references throughout the text must be correct according to the instruction of the journal.

 Minor editing of English language required

Author Response

Comments for reviewer 1

The manuscript describes the in vitro protective effect of L. acidophilus on particulate matter induced damage. The manuscript has several interesting features, and it should maker minor revision.

Introduction section: some related recent published articles must be added into the revised manuscript.

Response: Thank you for your comments. We have modified this manuscript according to your suggestion. Some related recent published articles have been added in the manuscript. Please check in Introduction section (LINE: 24-66).

Discussion section: It is recommended to merge Results and Discussion sections to better illustrate the mechanisms of action involved in this protective effect. Some subtitles of the results have no corresponding discussion in this manuscript.

Response: Thank you for your comments. We have added illustrations for each subtitle of the result in the discussion section. Please check in Discussion section (LINE: 279-352).

The citation of references: throughout the text must be correct according to the instruction of the journal.

Response: Thank you for your comments. The citations are checked and revised carefully according to the instruction of the foods journal.

Reviewer 2 Report

In this work, the authors investigate the Systematic investigation of the effect of Lactobacillus acidophilus TW01 on potential prevention of PM2.5-induced damage using a novel in vitro platform. The topic is interesting and the manuscript could have a potential from scientific point of view. However, there are major flaws that require the authors’ attention.

1.      Introduction Lines 23-30: I believe it is important that some more general information about PM2.5, should be added in the Introduction section (eg. their origin, frequency, when they become dangerous, etc). After all, these elements are highlighted also in the title.

2.      Line 31: Please present the lung-gut axis functionality in more details .

3.      Line 33: Give a few examples of PM-mediated inflammatory lung diseases.

4.      Line 35: Give a few more details about the subjects’ clinical state and their general health condition (e.g. difficulty in breathing, fatigue, etc) due to the impaired gut barrier function and reduced serum lipopolysaccharide (LPS) levels.

5.      Line 38: “…strategy for preventing PM damage.” Reference required.

6.      Line 40: I don’t believe that the bone mineral density is related to the topic of the manuscript. Please remove.

7.      Line 56: Give the full name of the laboratory.

8.      Have the strain isolates been tested for hemolytic activity and susceptibility to antibiotics? These are quite standard tests for strains intended for human consumption.

9.      Line 63: That is a great range 1x10^4-8 cfu/mL. Have the authors investigated all different concentrations (1x10^4, 1x10^5, 1x10^6, 1x10^7, 1x10^8 cfu/mL) each time? If not please add the phrase “…in different assays” at the end of this sentence.

10.   Line 64-65: It is not clear by this sentence if the bacteria were used alive or dead later on. Also, only concentrations 10^5-10^6 are shown here. This is confusing.

11.   Subsections 2.5, 2.6, 2.7 & 2.9: Appropriate references should be provided.

12.   Line 170: Please write clearly the dose of L. acidophilus TW01 applied.

13.   In general, results are well presented. Statistical significance should be incorporated in-text (where applicable).

14.   Most importantly, and despite having some interesting results, at its current form the discussion section is oriented mostly to describe results and no real discussion is provided (with the exception of lines 262-266). The connection and comparison of the authors’ findings to the existing literature (regarding similar final outcome, similar use of other strains or even other screening platforms, etc) should be promoted. At its current state, it is really scarce. This is also depicted on the very few references found in the literature section.

15.   Lines 297-302: There is no use for this paragraph here. The authors could combine this with the conclusions.

16.   Conclusions should be improved. Start with a summary of results (not too technical), and finalize with an outlook. More precise and comprehensive conclusions can be provided for readers. Future plans can also be showed. At its current state this section is rather poor, but the scientific picture given in the manuscript is more complex.

Minor editing.

Author Response

Comments for reviewer 2:

In this work, the authors investigate the Systematic investigation of the effect of Lactobacillus acidophilus TW01 on potential prevention of PM2.5-induced damage using a novel in vitro platform. The topic is interesting and the manuscript could have a potential from scientific point of view. However, there are major flaws that require the authors’ attention.

  1. Introduction Lines 23-30: I believe it is important that some more general information about PM2.5, should be added in the Introduction section (eg. their origin, frequency, when they become dangerous, etc). After all, these elements are highlighted also in the title.

Response: Thank you for your comment. We have modified this manuscript according to your suggestion. We have added some information about PM2.5: “Fine particulate matter, known as PM2.5 (PM diameter ≤ 2.5 µm), is an air pollutant that is a severe threat to human health worldwide [3]. PM exposure induces several respiratory diseases by prompting pulmonary inflammation and inducing oxidative stress [4,5]. These particles contain lots of dangerous chemical compounds, such as polycyclic aromatic hydrocarbons (PAHs), aromatic ketones and ethylene glycol, which are easily breathed into the lungs, contributing to many hazardous effects on the airway system [6]. Short-term exposure of PM2.5 highly increased the inflammatory factors in lung [7].” (Please see in LINE: 25-32)

  1. Line 31: Please present the lung-gut axis functionality in more details.

Response: Thank you for your comment. We have modified this manuscript according to your suggestion as followed:

“A study in children with cystic fibrosis observed that some bacteria were found in the intestinal tract before being identified in the respiratory tract, suggesting that micro aspiration might involve the intestinal microbes in the development of the respiratory tract microbiota [10, 11]. Accumulating evidence also demonstrated that the microbial composition in intestinal and respiratory tracts was closely related, and alteration of microbiota in intestinal or respiratory tracts could influence the other [11]. Additional-ly, many pulmonary diseases have been reported relating with a dysbiosis in the air-way and intestinal microbiota, indicating “lung–gut axis” influences each other [12].” (Please check in LINE: 36-43)

  1. Line 33: Give a few examples of PM-mediated inflammatory lung diseases.

Response: We revised this part of information as following: “The inflammatory lung diseases (such as asthma, pulmonary emphysema, and even lung cancer) [5,15] are associated with microbiota dysbiosis mediated by PM [16], resulting in impaired gut barrier function and reduced serum lipopolysaccharide (LPS) levels [17].” (Please see in LINE: 46-49)

  1. Line 35: Give a few more details about the subjects’ clinical state and their general health condition (e.g. difficulty in breathing, fatigue, etc) due to the impaired gut barrier function and reduced serum lipopolysaccharide (LPS) levels.

Response: Thank you for your comment. We have modified this in the manuscript of LINE: 46-49. “The inflammatory lung diseases (such as asthma, pulmonary emphysema, and even lung cancer) [5,15] are associated with microbiota dysbiosis mediated by PM [16], resulting in impaired gut barrier function and reduced serum lipopolysaccharide (LPS) levels [17].”

  1. Line 38: “…strategy for preventing PM damage.” Reference required.

Response: Thank you for your comment. We have added the references as following:

“Probiotics possess important health-promoting characteristics, including maintaining intestinal homeostasis and reducing inflammation, thus may provide a possible strategy for preventing PM damage [18,19].” (Please check in LINE: 50-52)

  1. Line 40: I don’t believe that the bone mineral density is related to the topic of the manuscript. Please remove.

Response: Thank you for your suggestion. We have removed the sentence according to your suggestion.

  1. Line 56: Give the full name of the laboratory.

Response: Thank you for your comments. We have modified this manuscript according to your suggestion. The lab name has been added. (Please see in LINE: 71-72) “L. plantarum MFM 30-3 and L. paracasei MFM 18 were isolated from Mongolian fermented milk (MFM) previously in Animal Product Lab, National Taiwan University [24].”

  1. Have the strain isolates been tested for hemolytic activity and susceptibility to antibiotics? These are quite standard tests for strains intended for human consumption.

Response: Thank you for your comments. L. acidophilus TW01 have been tested for the resistance of acid and bile salt. This study is mainly to screen potential strains with in vitro methods. The tested of L. acidophilus TW01 for hemolytic activity and susceptibility to antibiotics will be conducted in the near future.

  1. Line 63: That is a great range 1x10^4-8 cfu/mL. Have the authors investigated all different concentrations (1x10^4, 1x10^5, 1x10^6, 1x10^7, 1x10^8 cfu/mL) each time? If not please add the phrase “…in different assays” at the end of this sentence.

Response: Thank you for your comments. We have modified this manuscript according to your suggestion. The whole sentence changed as following:

“The lactic acid bacteria (LAB) were washed with 0.85% sodium chloride (NaCl, Sigma-Aldrich Chemical Co., St. Louis, MO, USA) solution or Dulbecco's phosphate-buffered saline (DPBS, Gibco, NY, USA) buffer twice before use and suspended in 0.85% NaCl or DPBS buffer at a concentration of 1 × 104-8 CFU/mL for use in different assays.” (Please check in LINE: 75-79 of revised manuscript)

  1. Line 64-65: It is not clear by this sentence if the bacteria were used alive or dead later on. Also, only concentrations 10^5-10^6 are shown here. This is confusing.

Response: Thank you for your advice. We have modified this manuscript according to your suggestion. “For heat-killed groups, the bacteria (1 × 105-6 CFU/mL density) were bathed at 80oC for 30 min.” Please check in LINE: 79-80.

  1. Subsections 2.5, 2.6, 2.7 & 2.9: Appropriate references should be provided.

Response: Thank you for your advice. We have modified this manuscript according to your suggestion. Please check it in LINE: 141, 150, 158, and 177.

  1. Line 170: Please write clearly the dose of L. acidophilus TW01 applied.

Response: Thank you for your suggestion. We have modified this manuscript according to your suggestion. The whole sentence changed in LINE: 203-206 as following:
“For the TBARS assay, L. acidophilus TW01 at 108 CFU/mL) demonstrated a better lipid peroxidation reduction (36.03±18.02 %) than L. plantarum MFM 30-3 (10.64± 6.21 %), L. paracasei MFM 18 (2.72± 2.72 %), and L. paracasei APL082 (13.81± 9.77 %) (Figure 1B)”

  1. In general, results are well presented. Statistical significance should be incorporated in-text (where applicable).

Response:

Thank you for your suggestion. We have added the p velum on the figures in manuscript according to your suggestion, please see Figure 2, Figure 3 and Figure 4 in the modified manuscript.

  1. Most importantly, and despite having some interesting results, at its current form the discussion section is oriented mostly to describe results and no real discussion is provided (with the exception of lines 262-266). The connection and comparison of the authors’ findings to the existing literature (regarding similar final outcome, similar use of other strains or even other screening platforms, etc) should be promoted. At its current state, it is really scarce. This is also depicted on the very few references found in the literature section.

Response: Thank you for your suggestion. We have modified discussion section in manuscript according to your suggestion, please check it in Discussion section of revised manuscript (LINE: 279-352).

  1. Lines 297-302: There is no use for this paragraph here. The authors could combine this with the conclusions.

Response: Thank you for your comments. We combined this paragraph with conclusion. Please check in LINE: 354-364. The conclusion changed as following:
“In the present study, we systematically screened probiotics on potential prevention of CSE -induced damage by a novel in-vitro platform. One strain, L. acidophilus TW01, possessing the potential protective effect on CSE-induced cell damage and death in vitro. The reduction of cell death by L. acidophilus TW01 might be involved in preventing the cell damage due to anti-oxidative and immune-regulatory effects, leading to reduced sub-G1 phase. L. acidophilus TW01 also showed the trend to recover the injured intestinal cell, suggesting gut barrier protecting effect. This study not only provides a novel screening platform for pre-screening potential probiotics in prevention of PM2.5-induced damage. It also illustrates the feasibility of this novel platform by successfully selecting a probiotic strain, L. acidophilus TW01, with the respiratory protective potential. Further in-vivo study is necessary to verify the in-vitro finding”

  1. Conclusions should be improved. Start with a summary of results (not too technical), and finalize with an outlook. More precise and comprehensive conclusions can be provided for readers. Future plans can also be showed. At its current state this section is rather poor, but the scientific picture given in the manuscript is more complex.

Response: Thank you for your suggestion. We have modified the section of conclusions in manuscript according to your suggestion, please check it in LINE: 354-364.

Reviewer 3 Report

Comments to Authors:

Title: Line 3: It will be more informative to use full form of PM in the title.

Abstract: Please add some more quantitative results in abstract.

Introduction:

1.       Line 35: check this, “reduced serum lipopolysaccharide (LPS) levels [9]” the LPS levels should be increased with respect to impaired gut barrier function.

2.       Add some more background in introduction to provide better background on the topic.

3.       Give rational behind the strain Lactobacillus acidophilus TW01 used in the present investigation.

Material and methods:

1.       Provide culture deposition numbers and or proper identify information by citing references.

2.       Correct the error for clear meaning: Briefly, 106-8 CFU/mL LAB were suspended in 75 µL of 85% NaCl solution and mixed with 25 µL of 2 mM ferrous chloride (FeCl2, Sigma- Aldrich Chemical Co.) and 50 µL methanol for 30 seconds.

3.       Line 73: specify the positive control.

4.       Line 80: specify the positive control for the activity.

5.       Line 94: what is the positive control? Specify.

6.       Provide detailed methods for cell culture and other experiments that how they are investigated for this study using LAB.

7.       Line 120: how many cells of LAB?

8.       Line 126: please don’t use the brand name of cigarettes. It will be more conflict free, if you use only cigarettes with properties like tar and nicotine and gram weight either whole or without paper and filter as appropriate to this study.

9.       Line 127: one cigarette after smoke or as such? Give clarity? If it is before burning or smoke, then change the title and other text in the manuscript. If it is after smoking i.e. ash, then the things will be different. Give clarity for this.

Results:

1.       Cells were not interfered in 517 nm OD analysis? Or the final solution was filtered? Correct methods if so. Or justify this.

2.       Specify the mean value significant differences in graph by using * as appropriate.

3.       What is PC in figure 2 A?

4.       Authors are sure about PM size in extract? How it was analysed?

5.       Figure 3A: the NC images are same for both the %? Please recheck the images to avoid duplicity.

Discussion:

1.       Provide the limitations of the present investigation.

2.       It will be nice to provide illustration about the concept that provide benefits to humans.

Need to improve

Author Response

Comments for reviewer 3:

Title: Line 3: It will be more informative to use full form of PM in the title.

Response: Thank you for your advice. We have modified this manuscript according to your suggestion. The whole sentence changed as following:

“Systematic investigation of the effect of Lactobacillus acidophilus TW01 on potential prevention of particulate matter (PM)2.5-induced damage using a novel in vitro platform.”

Abstract: Please add some more quantitative results in abstract.

Response: Thank you for your advice. We have revised the abstract with novel results as following: “Results showed that high dose of Lactobacillus acidophilus TW01 (1 × 108 CFU/mL) inhibited reactive oxygen species (ROS) production. This strain could also reduce respiratory epithelial cell death induced by cigarette smoke extraction (CSE), as well as promoting Caco-2 cell migration in 1 × 106 CFU/mL.” (Please see in LINE: 14-17 of revised manuscript)

Introduction:

  1. Line 35: check this, “reduced serum lipopolysaccharide (LPS) levels [9]” the LPS levels should be increased with respect to impaired gut barrier function.

Response: Thank you for your comment. This is a misunderstanding caused by a typo. We have modified this manuscript according to your suggestion. Please see in LINE: 46-49.

“The inflammatory lung diseases (such as asthma, pulmonary emphysema, and even lung cancer) [5,15] are associated with microbiota dysbiosis mediated by PM [16], resulting in impaired gut barrier function and reduced serum lipopolysaccharide (LPS) levels [17].”

  1. Add some more background in introduction to provide better background on the topic.

Response: Thank you for your advice. We have modified this manuscript according to your suggestion. Please check it in Introduction section (LINE: 24-66).

  1. Give rational behind the strain Lactobacillus acidophilus TW01 used in the present investigation. Please see in LINE: 54-55.

Response: “Lactobacillus acidophilus has been reported to stimulate and modulate the respiratory immune system in mice [20].”

Material and methods:

  1. Provide culture deposition numbers and or proper identify information by citing references.

Response: Thank you for your comments. We have added more information about the used cell lines as following:

“The human bronchial epithelial (HBEpiC, Cat:3210) cell line was purchased from Sci-enCell (CA, USA) and maintained in Bronchial Epithelial Cell Medium (BEpiCM, Cat:3211, ScienCell).” Please check in LINE: 124-126 of revised manuscript.

  1. Correct the error for clear meaning: Briefly, 106-8 CFU/mL LAB were suspended in 75 µL of 85% NaCl solution and mixed with 25 µL of 2 mM ferrous chloride (FeCl2, Sigma- Aldrich Chemical Co.) and 50 µL methanol for 30 seconds.

Response: Thank you for your comments. Here is a typo, we have corrected it. The sentence changed as following:

“Briefly, 106-8 CFU/mL LAB were suspended in 75 μL of 0.85% NaCl solution and mixed with 25 μL of 2 mM ferrous chloride (FeCl2, Sigma-Aldrich Chemical Co.) and 50 μL methanol for 30 seconds.” Please see in LINE: 94-96.

  1. Line 73: specify the positive control.

Response: Thank you for your comments. According to the protocol of DPPH assay, the butylated hydroxytoluene (BHT) is used as positive control (LINE: 88-90). We added the BHT standard curve in Supplementary Material.
(Please see the figure in attached file)

Figure S1. The butylated hydroxytoluene (BHT) standard curve for antioxidant assay (DPPH assay). The IC50 of BHT is 71.90 μg/mL. The data are presented as mean ± SD (n = 2).

  1. Line 80: specify the positive control for the activity.

Response: Thank you for your comments. According to the protocol of ferrozine ion chelating activity, the ethylenediaminetetraacetic acid (EDTA) is used as positive control (LINE: 98-101). We added the EDTA standard curve in Supplementary Material.
(Please see the figure in attached file)

Figure S2. Ethylenediaminetetraacetic acid (EDTA) chelating rate in ferrozine ion chelating activity assay. The IC50 of EDTA is 58.24 μg/mL. The data are presented as mean ± SD (n = 2).

  1. Line 94: what is the positive control? Specify.

Response: Thank you for your comments. In TBARS assay, we used 200 μg/mL of BHT as positive control and the result of BHT was added in Figure 1B. The sentence of positive control was added in LINE: 116-117 and please also check in Figure 1B.

  1. Provide detailed methods for cell culture and other experiments that how they are investigated for this study using LAB.

Response: Thank you for your comments. We have added more details to the experimental methods according to your suggestion (please see in LINE: 137-139)

“The HBEpiC cells were seeded (1.5 × 105 cells/well) into basolateral wells and incubated overnight at 37°C /5% CO2, and then treated with different CSE concentrations. L. aci-dophilus TW01 was added in the apical site for 24 h at the same time.”

  1. Line 120: how many cells of LAB?

Response: Thank you for your comments. We replenished the cell numbers of LAB. The whole sentence changed as following:
“The cells were then treated with 1 × 104-6 CFU/mL of L. acidophilus TW01 for 16 and 24 h. After treatment, the supernatant was removed, and non-adherent cells were washed off with DPBS buffer.” Please see in LINE: 143-146 of revised manuscript.

  1. Line 126: please don’t use the brand name of cigarettes. It will be more conflict free, if you use only cigarettes with properties like tar and nicotine and gram weight either whole or without paper and filter as appropriate to this study.

Response: Thank you for your comment. We make CSE according to the CSE preparation method mentioned by Cheng et. al. (2016). We have modified this manuscript according to your suggestion as following:
“The preparation method of CSE was according to Cheng et. al. [29]. Briefly, one cigarette (containing 0.9 mg nicotine and 10 mg tar) was lighted. The cigarette smoke was filtered into a 20 mL of the serum-free BEpiCM using a pump until the smoke was completely dissolved in a medium for extraction. The CSE solution was designated as 100% concentration and further diluted with BEpiCM for the experiments. This CSE solution was used no more than 30 min after being filtered through a 0.22 µM filter for each experiment.” (Please check in the section 2.6., LINE: 150-156)

  1. Line 127: one cigarette after smoke or as such? Give clarity? If it is before burning or smoke, then change the title and other text in the manuscript. If it is after smoking i.e. ash, then the things will be different. Give clarity for this.

Response: Thank you for your comments. We have modified this manuscript according to your suggestion. Please check in the section 2.6., LINE: 150-156.

Results:

  1. Cells were not interfered in 517 nm OD analysis? Or the final solution was filtered? Correct methods if so. Or justify this.

Response: Thank you for your comments. We centrifuged all samples for 3,000 rpm in 10 min to precipitate the LAB cells. We added this step in DPPH assay method, the subsection changed as following (LINE: 87-88 of revised manuscript):

“All samples centrifuged at 3,000 rpm for 10 min. The absorbance (OD) was measured at 517 nm using a microplate reader (BioTek Epoch, CA, USA)……”

  1. Specify the mean value significant differences in graph by using * as appropriate.

Response: Thank you for your comment. We defined that the data are presented as mean ±SEM (n = 3). The symbols indicate a significant difference compared to the negative (*p < 0.05, **p < 0.01, ***p < 0.001) and positive (#p < 0.05, ##p < 0.01 and ###p <0.001) controls. We have corrected the figure legend.

Figure 2. ……The data are presented as mean ±SEM (n = 3). The symbols indicate a significant difference compared to the negative (*p < 0.05, **p < 0.01, ***p < 0.001) and +LPS/positive (#p < 0.05, ##p < 0.01) group. (LINE: 229-231)

Figure 4. ……The data are presented as mean ±SD (n = 3). The symbols indicate a significant difference compared to the negative (*p < 0.05, **p < 0.01, ***p < 0.001). (LINE: 256-258)

  1. What is PC in figure 2 A?

Response: Thank you for your comment. The PC in figure 2A is 50 ng/mL LPS with Raw264.7 cells for 24 hours.

  1. Authors are sure about PM size in extract? How it was analysed?

Response: Thank you for your comment. We make CSE according to the CSE preparation method mentioned in Cheng et. al. [28]. The CSE solution have been filtered with 0.22 μm filter. All particles of CSE were under 0.22μm.

  1. Figure 3A: the NC images are same for both the %? Please recheck the images to avoid duplicity.

Response: Thank you for your comment. We treated 10% and 20% CSE with Caco-2 cell in the same dish and in the same time, these two dosages shared the same NC.

That’s why the NC images are same for both dosages.

Discussion:

  1. Provide the limitations of the present investigation.

Response: Thank you for your suggestion. We have modified the manuscript as following (LINE: 338-340 and 348-352):

“However, chemical components in PM2.5, originating from various sources with di-verse composition [44], are the key factor affecting adverse health effect [45].  The source of PM2.5 using in the study could significantly affect the results.”

And

“However, we noticed that high dose of L. acidophilus TW01 with high acid production could damage the single layer of attached caco-2 cells. In the intestinal barrier, tight junction proteins provide an important physical barrier [48]. The low cell viability might be due to the cytotoxic of lactic acid bacteria [49], resulting in a difficulty for Ca-co-2 cells to resist the organic acids produced by L. acidophilus TW01.”

  1. It will be nice to provide illustration about the concept that provide benefits to humans.

Response: Thank you for your comment. The last section of discussion was revised. Please check it in LINE: 287-293.

“Thus, in the present study, we designed an in-vitro platform for pre-selecting probiotics with a potential protective effect on PM-induced lung damage through determining the antioxidant ability, immune-regulatory effect, preventive effect of cigarette smoke-induced pulmonary injury, and enhancement of gut barrier, which includes the most of damage factors induced by PM2.5. A Caco-2 cell transwell system with HBEpiC cells was also developed to mimic the pro-biotic introducing pathway.”

Round 2

Reviewer 2 Report

Since no hemolytic activity nor antibiotics resistance was tested, I would recommend the authors to start the microorganisms description in materials and methods (l. 69) as follows: "Potential probiotic microorganisms..."

Minor editing 

Reviewer 3 Report

Authors revised the manuscript extensively, thus I hereby recommend to accept the said version.

Nil